# ACCEPTANCE: protocol for a feasibility study of a multicomponent physical activity intervention following treatment for cervical cancer

Nessa Millet [1], Hilary J McDermott,[1] Fehmidah Munir,[1] Charlotte L Edwardson,[2,3] Esther L Moss [4,5]

¹School of Sport, Exercise and Health Sciences, Loughborough University, Loughborough, UK
²Diabetes Research Centre, University of Leicester, Leicester, UK
³NIHR Leicester Biomedical Research Centre, Leicester, Leicestershire, UK
⁴Leicester Cancer Research Centre, University of Leicester, Leicester, Leicestershire, UK
⁵Department of Gynaecological Oncology, University Hospitals of Leicester, Leicester, UK

**Correspondence to**
Dr Esther L Moss;
em321@le.ac.uk

## ABSTRACT

**Introduction** Cervical cancer treatment can have life changing sequelae and be associated with poor short-term and long-term quality of life. Physical activity (PA; that is, bodily movement) is known to improve health outcomes and quality of life for cancer survivors, both physically and psychologically. To date, no interventions to increase PA following cervical cancer have been evaluated. This study aims to (1) determine the feasibility of conducting a PA intervention after cervical cancer and (2) to explore the acceptability of the programme and evaluation measures.

**Methods and analysis** The design is a pre study and post study design. Thirty participants aged between 18 and 60 years from the Midlands region, UK, who have completed primary treatment for cervical cancer at least 6 months previously and do not meet the national PA guidelines will be recruited. Identification of potential participants will take place through the University Hospitals of Leicester National Health Service (NHS) Trust. Participants will receive an intervention focused on increasing PA through the provision of education, action planning, goal setting, problem solving and self-monitoring of PA behaviour, particularly steps per day. Device assessed PA and questionnaires will be completed at baseline, week 6, week 12 and week 24. Feasibility will be assessed in terms of recruitment, retention, attrition, completion of measures and intervention compliance, for which specific feasibility criteria have been established. The process evaluation will explore the experiences and acceptability of the intervention components and evaluation measures.

**Ethics and dissemination** Ethical approval has been granted by the West of Scotland Research Ethics Committee 1 for this study. Results will inform intervention refinement for the design of a definitive pilot trial. These results will be disseminated via peer-reviewed publications and international conferences while input from a patient and public involvement (PPI) group will inform effective ways to circulate results among the wider community.

**Trial registration number** ISRCTN16349793, Registered 30 September 2020.

## INTRODUCTION

Advancements in the clinical management of cervical cancer have resulted in improved survival,[1,2] leading to a change in focus to treatment-related sequelae and survivorship issues, to reduce iatrogenic morbidity and to improve patients' long-term health.[3] Although many patients diagnosed with a cervical cancer have early stage disease (stage IA), which can be successfully treated with local excision, 64% of women have at least stage IB disease at diagnosis.[4] As a result, many patients will need to undergo radical surgery and/or chemoradiotherapy, that consequentially leads to the loss of fertility and an early menopause.[3]

The long-term effects of radiotherapy on the vagina,[5] bladder[6] and bowel can occur in between 30% and 40% of patients and can result in distressing symptoms, urinary/faecal such as incontinence[7] and tenesmus.[8] It can also lead to permanent physical changes, vaginal shortening/stenosis and reduced clitoral sensitivity, thereby physically limiting the ability to have and enjoy intercourse.[9] Chronic fatigue and pelvic insufficiency fractures are also frequently reported following radiotherapy treatment for cervical cancer, the symptoms of which may not diminish with time,[10] and can have a negative impact on daily functions and social quality of life.[11]

### Strengths and limitations of this study

► The integration of behaviour change theory and insight from the target population to inform the development of practical strategies ensures an appropriate and user-friendly intervention.
► A mixed methods process evaluation will provide a great depth of understanding relating to the experiences and acceptability of the intervention and will enhance subsequent refinement where necessary.
► As this is a feasibility study, the analysis will not report statistically powered differences in outcomes (e.g., device-assessed PA).

There is also substantial evidence of the devastating psychological consequences of a cervical cancer diagnosis. A high proportion of patients experience long-term psychological distress following their diagnosis,[12] which can impact on their social, work and functional well-being. The psychological impact is more pronounced in younger patients and those who receive chemoradiotherapy,[13] with reported lower levels of self-confidence and a more negative body image compared with those who undergo surgery alone.[14] Hence, a greater focus on cervical cancer survivorship is warranted to support patients who are living alongside the physical, psychological and social impacts of cervical cancer treatments.

Identifying and implementing ways to manage treatment side-effects can be challenging, given the individualistic and complex nature of the survivorship experience.[15] Moreover, concerns have been raised regarding the knowledge provision of treatment-related morbidity and clinical support services.[16] Physical activity (PA), defined as 'any bodily movement produced by skeletal muscles that results in energy expenditure' ([17] p126) has been shown to be a safe avenue to improve quality of life following cancer treatment,[18] with qualitative data indicating that PA can specifically improve physical, psychological, social and spiritual domains.[19] However, PA levels are reportedly low and remain low up to 3 years following treatment for a gynaecological malignancy.[20] In particular, cervical cancer survivors meet recommended PA guidelines less frequently than patients following other cancer types.[21] Exercise recommendation for cancer survivors have been suggested[22]; however, these do not target the specific PA barriers faced after cervical cancer, thus supporting the need for tailored PA promotion in this population.

To date, there has been little research focus on promoting PA among those treated for cervical cancer. This is highlighted by cervical cancer cases being under-represented in meta-analysis studies of gynaecological cancer.[18] Literature based in the broader context of gynaecological cancer and PA does provide some degree of insight regarding PA preferences and feasibility of different types of PA. Such studies suggest that PA which: incorporates home-based walking,[23–25] and gym activities[25]; is graded and flexible[24]; is combined with a form of support or counselling[24]; includes social interaction with peers,[26] tends to be favourable among survivors of gynaecological cancers. There is growing support for technology-based PA promotion among cancer survivors,[27] which can place less burden on participants, and can be easily tailored to the individual. However, the PA barriers faced after cervical cancer may differ to other gynaecological cancers due to variations in patient characteristics (e.g., age) and treatment side effects. Therefore, it is important to consider the specific needs of cervical cancer survivors during intervention design and implementation.

To the best of our knowledge, an intervention which promotes PA participation following treatment for cervical cancer has not been evaluated. Therefore, we aim to test the feasibility and acceptability of conducting and evaluating a multicomponent intervention programme, informed by relevant theories of behaviour change, to increase PA levels following treatment for cervical cancer. This intervention has been named: Acceptability in Cervical Cancer of an Exercise-based Programme delivered Through An oNline Community Environment (ACCEPTANCE). This intervention was developed using the intervention mapping protocol.[28]

## Aim

The overall aim of this study is to assess the feasibility and acceptability of a PA intervention following treatment for cervical cancer.

## Objectives

1. To determine participant recruitment, retention and attrition rates.
2. To review descriptive data to determine the feasibility of the inclusion criteria.
3. To determine compliance and completeness of evaluation measures at all time points.
4. To report the feasibility of delivering the intervention and evaluation measures.
5. To explore participant compliance with the intervention.
6. To obtain views on the acceptability of the intervention and evaluation measures.
7. To describe device-assessed PA levels and questionnaire data outcomes of interest.

## METHODS AND ANALYSIS
### Study design

This is a single-arm pre study and post study to assess the feasibility and acceptability of conducting and evaluating a PA intervention following treatment for cervical cancer. A process evaluation will also be conducted throughout the study.

### Study setting

This study will be conducted through the University Hospitals of Leicester (UHL) National Health Service Trust, in collaboration with Loughborough University and the University of Leicester.

### Patient and public involvement (PPI)

PPI consisted of two groups of women treated for cervical cancer within the previous 10 years. PPI input has informed the development of the study since its inception. Insights and feedback gathered from one to one and group discussions with PPI members enabled development of the research question, assessment of the potential burden on participants and informed the intervention content to ensure that intervention materials were user friendly (table 1).[29] Continuation of PPI involvement will inform the intervention delivery and dissemination of results to the wider community as appropriate.

**Table 1** Patient and public involvement (PPI) involvement and engagement in research activities

| Research stage | Involvement | Mode |
|---|---|---|
| Research conception and design | Establishment of unmet needs after cervical cancer | In person group discussion PPI+RT |
| | Sharing personal experience of recovery after cervical cancer | One to one conversation PPI+SC |
| Recruitment for feasibility trial | PPI members provide feedback on recruitment poster | Email correspondence PPI+SC |
| | PPI members share study details via word of mouth or poster on online survivorship group | Email correspondence PPI+SC |
| Intervention design | PPI group to provide feedback on the feasibility of the intervention timeline | Group discussion between PPI+SC; use of questionnaire to elicit feedback |
| | PPI members to complete evaluation questionnaire and provide feedback on length and language use | One to one in person conversation PPI+SC |
| | PPI group to evaluate intervention materials (diary, education and *fitbit* guide) | Email correspondence PPI+SC |
| | PPI group to complete intervention launch session and provide feedback | Virtual group discussion PPI+SC |
| Programme delivery | PPI group to feedback on progress of intervention at mid-intervention delivery | Virtual group discussion PPI+SC |
| Dissemination of results | PPI group to feedback on appropriate strategies to disseminate results to the wider public (including patients and survivors) | Virtual group discussion PPI+SC |

RT, research team; SC, study coordinator.

## Participants and recruitment

Participants between the ages of 18 and 60 years who have undergone treatment for cervical cancer at least 6 months previously with curative intent (either surgery, chemoradiotherapy or both modalities), who do not meet the national PA guidelines (150-300 min of moderate to vigorous intensity PA per week),[30] who can gain permission from their gynaecologist/oncologist to take part and are proficient in speaking English will be recruited. The age range of 18–60 years was chosen to reflect the Chief Medical Officer's PA adult guidelines and to optimise engagement and adherence to a technology-based PA programme. Exclusion criteria are benign or premalignant (cervical interepithelial neoplasia) disease of the cervix; clinical/radiological evidence of disseminated malignancy; pregnancy or breast feeding; performance status ≥3; comorbidity that in the opinion of the patient's supervising gynaecologist/oncologist would preclude the patient from meeting the study requirements.

Initial database screening will take place by the clinical gynaecological oncology team to identify candidates who are at an appropriate age and stage in their treatment journey. Potentially eligible participants will be approached through their clinical team; recruitment posters will be placed in the gynaecological and oncology outpatient departments at UHL; and study details (overview of research, inclusion criteria and relevant contact details) will be posted on social media platforms. Local cervical cancer support groups will also be contacted and given information on the study to raise awareness of the research. Screening will take place via a telephone call with a member of the research team. The Scottish Physical Activity Questionnaire[31] will be used to determine PA levels. Participants will be asked how many minutes per week they spend being active at a moderate intensity in the context of the last 7 days. Consent procedures will take place either virtually via video/telephone call or in person at UHL.

## Sample size

This study will aim to recruit approximately 30 participants. This sample size was chosen as it is a realistic target when considering cervical cancer incidence both in the UK and at UHL where recruitment will take place over the given time period. In line with recommendations from the National Institute for Health Research, a sample size of 30 is appropriate to answer the questions posed by a feasibility trial.[32] The feasibility of lifestyle and PA interventions in endometrial and breast cancer survivors have successfully employed similar sample sizes.[27 33]

## Intervention

The aim of the multicomponent intervention is to increase PA levels of the target population, specifically, individual and group-based walking over 12 consecutive weeks (figure 1). Table 2 provides a description of each intervention component while figure 2 outlines the Logic Model. Walking has been identified as a common PA preference among gynaecological cancer survivors,[34] particularly for those who do not meet the recommended PA guidelines.[35] The intervention components are underpinned by social cognitive theory,[36] and are informed by the health belief model,[37] and theories of self-regulation.[28] Self-monitoring behaviour is strongly linked with successful

**DAY ONE: Intervention launch**

- Goal Setting
- Education
- Barrier Identification
- Problem Solving

↓

- Wear Self- monitoring activity monitor daily
- Daily and weekly Intervention diary
- 6 x fortnightly health coaching sessions
- Messaging via online community between programme groups
- Participant led group walking

↓

**WEEK 12**

**Figure 1** Intervention programme flow chart.

PA behaviour change, particularly when combined with at least one other self-regulatory component.[38] The intervention includes education provision, problem-solving and goal setting in relation to increasing PA levels after treatment for cervical cancer. Self-monitoring of PA behaviour will be facilitated by providing participants with an intervention diary to complete daily and weekly, fortnightly online health coaching via video or telephone call administered by the study coordinator and a consumer PA monitor (*fitbit inspire 2*) to receive real-time feedback on their activity levels and prompts to be more active and to review goals. Throughout, peer support and group walking among participants will be encouraged via a messaging platform, which will allow participants to maintain contact and organise group walking sessions. The study coordinator will suggest appropriate walk locations convenient to each group.

## Measures
All PA and health measures will be assessed at baseline, week 12 and 3 months after completion (week 24). Previous literature suggests that psychological measures may take longer to change than physical measures (e.g., menopausal symptoms or PA) and therefore to reduce participant burden, only these will be taken at week 6 (table 3).

## PA and health measures
An accelerometer (GENEActiv, Activinsights) and sleep log will be administered at all evaluation time points to measure PA. Participants will be asked to wear the accelerometer on their non-dominant wrist 24/7 for 8 days at each time point. Valid data will be at least 3 days >16 hours of accelerometer data collected.[39] Device-assessed PA will be the primary research outcome measure of interest in the subsequent pilot study and definitive main trial.

Participants' belief in their ability to successfully complete incremental 5–50 min periods of walking at a moderate/brisk pace will be assessed via the self-efficacy for walking scale.[40] Motivation for PA will be assessed using the Behavioural Regulation in Exercise Questionnaire.[41] This is a 23-item scale which assesses motivation on a self-determination continuum. Enjoyment of PA will also be assessed via the original 18-item PACES scale.[42]

In relation to well-being, menopausal symptoms, such as hot flushes and sweating will be investigated using the Menopausal Rating Scale.[43] Quality of life will be assessed using the EORTC QLQ-30, which is commonly used in gynaecological cancer populations and includes five functional scales, three symptom scales, a global health scale and a quality of life scale. The Hospital Anxiety and Depression scale (HADS),[44] will be administered to measure symptoms of anxiety and depression. Fatigue will be assessed using the Multi-dimensional Fatigue Symptom Inventory (MFSI-SF),[45] a measure which has been previously used to assess fatigue in a gynaecological cancer population.[23]

## Evaluation-related feasibility measures
Participant recruitment rates will be monitored by recording the number of identified potential participants, the number of those who express an interest in taking part, the number of those who are eligible and not eligible along with the reasons for ineligibility and the number who consent to participate. Retention and attrition rates will be monitored by recording the number of participants who withdraw from the study and the number who do not attend follow-up. Where possible, reasons for withdrawal will also be reported. The number of participants who complete the PA and health measures at each time point will be summarised.

## Process evaluation
The process evaluation will be conducted to explore participant experiences of the study and intervention as well as compliance with the intervention components, to identify facilitators and barriers to increasing PA and any suggestions for improvement and refinement. Compliance with intervention components will be reviewed throughout the intervention. Data will be presented on launch attendance rates, the number of diary entries completed, number of daily step counts recorded to reflect compliance of wearing the PA monitor, number of health coaching sessions attended, the number of participants who post on the online forum and engage in group

**Table 2** Description of intervention components

| Intervention component | Description |
|---|---|
| *Intervention launch* | **Education** (interactive session facilitated by the study coordinator, delivered virtually)<br>► What is meant by PA?<br>► What kinds of activities count as PA?<br>► What does moderate intensity PA feel like?<br>► What benefits are associated with PA?<br>► Knowledge of PA recommendations<br>► How many steps per day should you aim for?<br>**Barrier Identification**<br>► Is there anything that might stop you from being physically active?<br>► How can you overcome these challenges?<br>**Goal Setting**<br>► Introduction to SMART goals<br>► Using PA monitor to set goals related to steps and intensity<br>*Each section divided by guided 'stand up and stretch' prompts |
| *Self-monitoring of PA* | ► The PA monitor will provide participants with feedback on their activity levels via: number of steps taken by participants and the number of minutes that participants spend in a moderate intensity activity each day |
| *Prompts* | ► Prompts offered by PA monitor to encourage PA<br>► Prompts to update participant on their progress in reaching their daily goal |
| *Intervention diary (monitoring of well-being)* | **Daily entries**<br>► Input daily step count as shown on PA monitor<br>► Rate mood out of 10<br>**Weekly entries** |
| | ► Rate the following symptoms on a traffic light system: energy levels, anxiety levels, body confidence, physical pain, bladder issues |
| *Health coaching* | ► Revision of previous goal(s) and the outcome (successful/ unsuccessful); attribute reasons for this outcome<br>► Identification of challenges and facilitators regarding previous goal<br>► Revision of steps, intensity and well-being over the previous 2 weeks<br>► Facilitated goal setting and identification of potential challenges for the following 2 weeks |
| *Group messaging* | ► To be used as a platform to schedule group walks (organising a time and a place) with the aim of one group walk per week. Standardised communication prompts to be offered by the study coordinator |
| *Online community* | ► Participants will be encouraged to use the study-specific community on the PA monitor application, accessed via smart phone where personal PA insights and statistics can be shared with other participants |

PA, physical activity.

messaging, the number of participants who take part in group walks and the frequency of these.

Questionnaires will be administered to understand participants' experience of attending the intervention launch, of using the PA monitor and the group messaging. A semi-structured interview will also be conducted by the study coordinator to further explore perceptions of the intervention and the evaluation measures, perceived benefits of participation, maintenance of the intervention components and PA levels. Interview data will be analysed using Template Analysis,[46] employing a deductive coding framework to inform study refinement. To ensure a balanced insight and that subsequent meaningful refinement can be implemented, some transcripts will be coded by an independent researcher who did not contribute to the intervention design.

A researcher log will be completed by the study coordinator where a record of interactions with participants (e.g., email conversations/telephone conversations) will be kept throughout the duration of the study. Feedback will be broadly divided into four categories: evaluation of intervention components; experience of evaluation measures; alterations to future intervention implementation; barriers to PA and reasons for discontinuation (where relevant).

## Data collection

The accelerometer and the quantitative measures will be posted to participants. Process evaluation questionnaires will be completed online throughout the intervention. The evaluation interview will be offered after the week-24 time point to all participants (including those who have withdrawn) either in person or via video call.

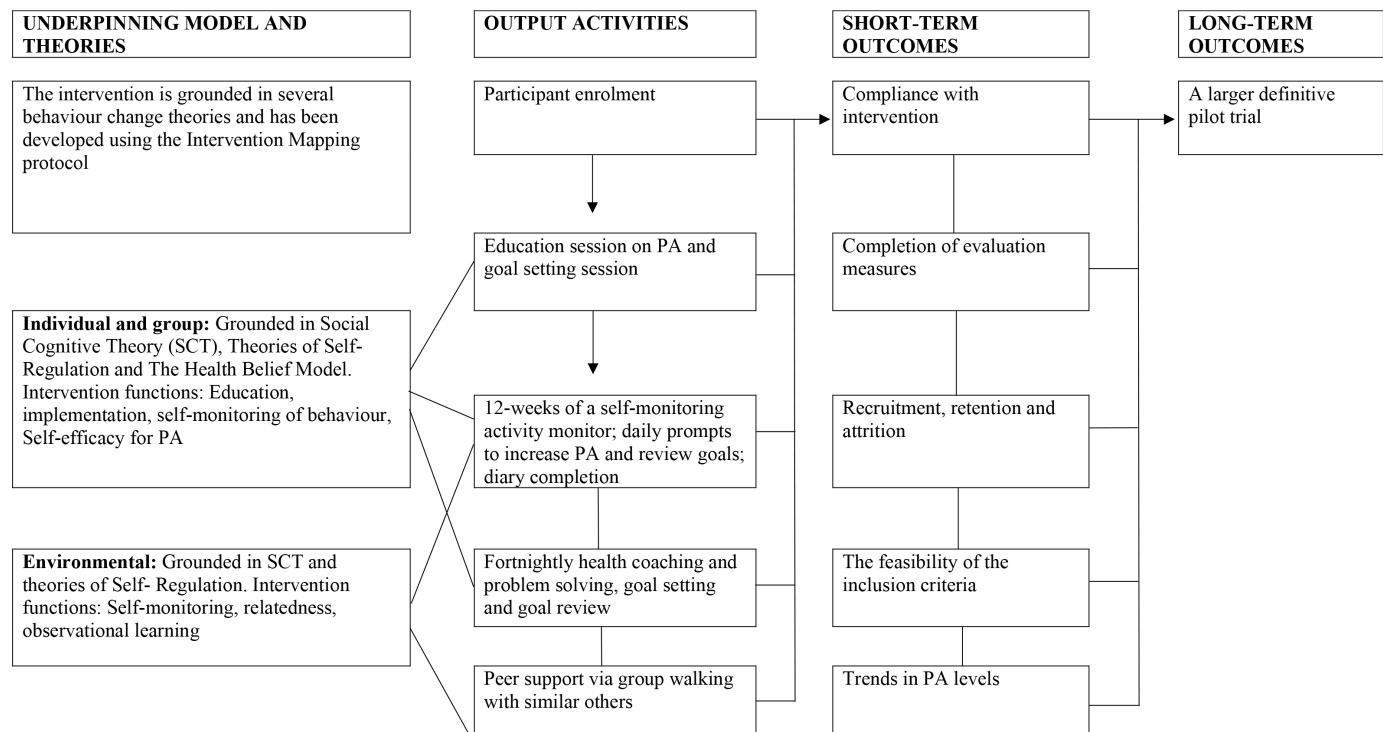

**Figure 2** Logic model for the ACCEPTANCE trial. PA, physical activity; SCT, social cognitive theory.

### Statistical analysis

As this is a feasibility study, main analyses will use descriptive statistics. Data will be analysed using *IBM SPSS statistics V.25.*

### The feasibility of inclusion criteria

We have suggested that if an eligibility criterion accounts for 40% of ineligibility, it will be reported.

### The feasibility of study recruitment and retention

A priori, we have defined a recruitment success rate as recruitment of 30 participants in the first 3 months or that 75% of those identified as eligible are recruited, and a successful retention rate at week-24 as 70%.

### The feasibility of and compliance with intervention components

Attendance rates at the intervention launch, engagement with the online forum, frequency of wearing the PA monitor, frequency of group walking sessions, compliance with diary completion and attendance rates at health coaching sessions will all be summarised. The threshold of feasibility for these rates will be 70% of participants complying, respectively.

### The feasibility of the evaluation measures

Compliance with evaluation measures will be deemed successful if no more than 20% of participants fail to provide questionnaire and accelerometer data at both baseline and week-24 follow-up.

**Table 3** PA and health outcomes, outcome measures and measurement time points

| Measures | Assessment tool | Baseline | Week 6 | Week 12 | Week 24 |
|---|---|---|---|---|---|
| Device-assessed PA | Accelerometer | x | x | x | x |
| Menopausal symptoms | MRS | x | x | x | x |
| Quality of life | EORTC-QLQ-30 | x | | x | x |
| Depression and anxiety | HADS | x | | x | x |
| Fatigue | MSFI-SF | x | | x | x |
| Walking self-efficacy | Self-efficacy for Walking scale | x | | x | x |
| Motivation for exercise | BREQ-3 | x | | x | x |
| Enjoyment of PA | PACES | x | | x | x |

BREQ-3, Behavioural Regulation in Exercise Questionnaire; EORTC-QLQ-30, European Organisation of Research and Treatment of Cancer Core Quality of Life Questionnaire ; HADS, Hospital Anxiety and Depression Scale; MFSI-SF, Multi-dimensional Fatigue Symptom Inventory; MRS, Menopausal Rating Scale; MSFI-SF, Multidimensional Fatigue Syndrome Inventory- Short Form; PA, physical activity; PACES, Physical Activity Enjoyment Scale .

### PA data and questionnaire outcomes

Data from the accelerometer will be cleaned and processed in GGIR using *R programming* and analysed using *IBM SPSS statistics V.25*. Means and standard deviations (SDs) will be used to describe the PA from the accelerometer at each evaluation time point. Constructs measured via questionnaire will be analysed by standard scoring procedures. Questionnaire outcomes of interest will be reported on means (or medians where necessary) and SDs.

### Ethics and dissemination

Ethical approval has been granted by the West of Scotland Research Ethics Committee 1 for this study. Results will primarily be used to refine intervention components and to inform a larger full-scale trial, if appropriate. Results from the study will be shared in peer-reviewed journals, presented at international conferences and will be shared with the UHL and Leicester Hospital's charity.

## DISCUSSION

This article describes the protocol for a study testing the feasibility and acceptability of conducting and evaluating a multicomponent PA intervention following treatment for cervical cancer. Regular PA has been shown to have physical and psychological benefits and to be safe after cancer treatment.[47] PA can also alleviate a range of symptoms experienced as a result of cancer treatment .[23] PA levels appear to be low following cervical cancer[21] and this may be partly due to the lack of a specific intervention that takes into consideration the particular side-effects that can occur, in particular pelvic pain and incontinence, that may not be such an issue in other cancer types.

The study protocol details a 12-week intervention programme to promote walking and PA, with evaluation measures taken at baseline, 6 weeks, 12 weeks and 24 weeks. The process evaluation is a mix of questionnaires and a qualitative interview. The intervention was designed using the intervention mapping protocol.[28] Development followed a systematic six-step framework, which prioritises input from the target population as well as integrating behaviour change theories (BCTs) and constructs to underpin change within the intervention, informed by relevant literature.[48] Interventions underpinned by relevant BCTs have been found to be more successful in implementing change and in identifying more specific evaluation criteria.[47]

The current PA guidelines offered by the Government Chief Medical Officers do not specify recommendations for cervical cancer survivors.[30] However, using a development framework that prioritises insights and experiences from the target population is a great strength of this intervention, as it allowed direct and effective tailoring of the intervention components despite specific recommendations not being available. Hence, the aerobic PA recommendations were deemed an appropriate target for women treated for cervical cancer.[29] Additionally, the criterion to include women who are at least 6 months post treatment was also a decision informed by the need's assessment, with the time between 3 and 9 months post treatment being identified as critical for intervention and for positive change.[49] A limitation is that the protocol does not incorporate strength training, and therefore does not reflect the current PA guidelines for after cancer. However, as PA levels are generally low following cervical cancer treatment, it seemed appropriate to promote one PA modality which was realistically achievable (i.e., walking) to enhance adherence to the intervention.

This study will add to the current debate regarding the role of technology in promoting PA after cancer. The intervention uses a PA monitor and its associated online community and a messaging platform. Previous research supports the use of such e-health technologies in combination with health coaching over a shorter duration (4-week programme).[50] However, it has been suggested that further investigation is needed to test the acceptability of such technologies in more diverse populations,[51] to explore optimal intervention length and to question the maintenance of PA behaviour change.[50] Although a limitation of this protocol is that statistically relevant changes in PA between time points will not be reported, trends in PA levels will provide data inferring the feasibility of promoting and maintaining PA behaviours. The process evaluation will also provide valuable insight regarding the acceptability of the components.

## CONCLUSION

We described the protocol of a study which tests the feasibility and acceptability of delivering and evaluating a multicomponent PA intervention for women following treatment for cervical cancer. A process evaluation will provide insight into whether the intervention components and evaluation measures are accepted by participants. Findings from the study will inform intervention refinement in preparation for a full-scale pilot trial.

**Acknowledgements** The authors are very grateful to the members of the patient and public involvement group for their contributions and insights throughout the intervention development stages.

**Contributors** All authors contributed to the study conception and design and protocol preparation. The first draft of the manuscript was written by NM and ELM. CLE, FM and HJM significantly revised the manuscript. All authors read and approved the final manuscript.

**Funding** This research is supported by the National Institute for Health Research (NIHR) Leicester Biomedical Research Centre, which is a partnership between University Hospitals of Leicester NHS Trust, Loughborough University and the University of Leicester, UK. Support. Funding was provided by the Economic, Social and Research Council (ES/P000711/1), Loughborough university (ES/P000711/1), and the Leicester Hospital's charity, University Hospitals of Leicester (Q843).

**Competing interests** ELM has received research grants from Intuitive Surgical and Hope Against Cancer for unrelated work. ELM has received lecture fees for GlaxoSmithKlein and has served on the clinical advisory boards for Inivata and GlaxoSmithKlein.

**Patient and public involvement** Patients and/or the public were involved in the design, or conduct, or reporting, or dissemination plans of this research. Refer to the Methods section for further details.

**Patient consent for publication** Not applicable.

**Ethics approval** Ethical approval was given by the West of Scotland Research Ethics Committee (20/WS/0062).

**Provenance and peer review** Not commissioned; externally peer reviewed.

**Data availability statement** The submission is a study protocol and therefore data sharing is not applicable as no datasets have been generated and/or analysed for this study.

**Open access** This is an open access article distributed in accordance with the Creative Commons Attribution 4.0 Unported (CC BY 4.0) license, which permits others to copy, redistribute, remix, transform and build upon this work for any purpose, provided the original work is properly cited, a link to the licence is given, and indication of whether changes were made. See: https://creativecommons.org/licenses/by/4.0/.

**ORCID iDs**
Nessa Millet http://orcid.org/0000-0001-5468-5197
Esther L Moss http://orcid.org/0000-0002-2650-0172

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
