## [Reviewer comments · BMJ Open]

ARTICLE DETAILS

TITLE (PROVISIONAL)	ACCEPTANCE: Protocol for a feasibility study of a multicomponent physical activity intervention following treatment for cervical cancer
AUTHORS	Millet, Nessa; McDermott, Hilary J; Munir, Fehmidah; Edwardson, Charlotte; Moss, Esther

VERSION 1 – REVIEW

REVIEWER	Edward Stanhope Staffordshire University
REVIEW RETURNED	24-May-2021

GENERAL COMMENTS	The dates of the study should be included in the manuscript, as per the notes from the editors: instructions for reviewers of study protocols Is the abstract accurate, balanced and complete? A definition of physical activity would be desirable More clearly report what is considered to be the primary outcome(s) of interest (feasibility) and what is considered secondary outcome(s) of interest (QoL, anxiety and depression, fatigue etc). Report how feasibility will be assessed, at what time points and how it will be analysed. Specifically, what are the criteria for concluding feasibility? Are the methods described sufficiently to allow the study to be repeated? Participants and recruitment: More detailed is needed on the 'screening to determine potential participants PA levels will take place via a telephone call with a member of the research team'. What counts as physical activity and over what time period is this being considered? Intervention: It would be useful to include 'number of minutes that a participant spends in a moderate intensity activity each day' in the daily intervention diary. A number of the weekly diary entries are likely to fluctuate throughout the week/day, what is the rationale for only collecting them weekly? Who will be administering the health coaching sessions? Are these also delivered virtually? Measures: Clarity is needed for the following statement: '...whilst those measures expected to change, based on literature, will be taken at week 6'. It is unclear what the author means by this; are all items not expected to change? It would be beneficial to discuss the outcomes as primary outcomes relating to feasibility and acceptance and secondary outcomes relating to PA and health.
--

	Clarity is needed for the following statement: 'The completion rates of the PA and health measures at each time point will be summarised'. Do the authors mean the number of participants that data is collected for, and the completeness of data or something else? Statistical analysis: 'If an eligibility criterion accounts for 40% ineligibility, it will be investigated further' – how? Provide the specific investigation planned. What is the rationale for a recruitment rate of 30 participants in the first 3-months as the threshold for feasibility? Also, what is meant by '75% of those identified are recruited'? 75% of eligible participants or 75% of potential participants approached? In regard to feasibility of and compliance with intervention components, what threshold will be used to conclude feasibility?
--	--

REVIEWER	Gianina Postavaru Bishop Grosseteste College, Applied Social Sciences
REVIEW RETURNED	03-Jul-2021

GENERAL COMMENTS	I welcome the opportunity to review this interesting article. I have some comments which might be helpful to improve it. Abstract, Ethics and Dissemination: The dissemination statement needs to align more clearly with the PPI section, where it is stated that the PPI will involve in 'dissemination of results to the wider community as appropriate'. Please outline any planned outputs resulting from the engagement (not just involvement) with the PPI/E. Introduction: This section could provide more clarity in relation to the PA guidelines for cancer survivors and the lack of recommendations for cervical cancer survivors, as stated later in the Discussion. It feels that this addition would give a clearer context for this piece of research being timely and much needed. Lines 41-48: please review punctuation. Methods and Analysis, Patient and Public Involvement (PPI): Collaboration with PPI/E is gold standard in health research; therefore, authors are to be congratulated for the effort. However, the research team does not clearly articulate who was part of the PPI group (patients, carers, members of support groups, members of specific communities, people living beyond cancer etc), mainly that the PPI members were involved in the assessment of the 'potential burden on participants' and had an active role in informing the intervention content. How were these 'discussions' managed, and in what ways was feedback collected and integrated into the intervention content? Please indicate if a PPI/E protocol exists and where it can be accessed. It would be useful to learn whether a PPI/E plan has been designed to outline the 'involvement' and 'engagement' elements of this collaboration throughout different stages of the project and expected outputs. I would suggest a table describing the different activities that the PPI will contribute to (e.g. how they might get involved in different aspects of this project, their role in commenting on and developing the patient facing documents or other research materials, if there is any involvement in data collection or analysis as user researchers etc). Authors mention that the PPI/E will be involved in the dissemination of results to the wider community. Authors do state that results will
---

	be shared with the University Hospitals of Leicester and Leicester Hospital's charity. Please give examples of planned activities, if any (e.g. science festivals open to the public or debates on research open days where members of the public are invited, awareness of research through media, dissemination to members of the public etc). In addition, has the research team considered involving the PPI/E in the Equality and Health Inequalities Impact Assessment of the intervention (e.g. addressing potential inequalities due to race and ethnicity, learning or physical disabilities, digital exclusion etc)? Participants and recruitment: Please provide a justification for the upper age limit. The feasibility of inclusion criteria: Please capitalize the first letter of the first word.
--	--

REVIEWER	Natalie Vear The University of Queensland - Saint Lucia Campus
REVIEW RETURNED	09-Jul-2021

GENERAL COMMENTS	Reviewer's Comments 09/07/2021 Overall Comments: The authors present a protocol paper outlining the design and methodology of a feasibility study investigating the use of a multicomponent physical activity intervention in those following treatment for cervical cancer. Overall, this paper provides the details of an interesting and much-needed intervention for those living with and beyond cervical cancer. The following general comments refer to referencing issues and suggested formatting changes throughout the paper. The authors need to review their references and ensure relevant, peer-reviewed references are cited. They must also ensure that all statements referring to previous research or work is referenced accordingly. I would recommend the authors carefully consider sentence length in some of their sections (especially the introduction). This will assist with the flow of their writing and reader comprehension. All figures and tables could have acronyms spelt out in full in the Figure/Table header/footer to enable ease of reading if being viewed independently. Specific comments are provided below according to article section, page number and line number/s: Specific Comments: Abstract: Page 4, Line 22: Spell "UK" in full Introduction: Page 6, Lines 6-9: This statement should be supported by peer-reviewed literature (or specific government reports) relevant to changes in cervical cancer survival rates Page 6, Lines 9-13: Reference needed Page 6, Lines 29-36: Reference/s needed to support these side effects Page 6, Lines 54-57: I suggest rewording "reportedly more greatly
---

	affected” Page 7, Line 17: This reference does not refer in any way to “provision of treatment-related morbidity and clinical support services” for cervical cancer. Please replace reference Page 7, Lines 36-41: References needed Methods and Analysis: Page 9, Lines 25-37: Do you have any published work referring to this previous PPI work? If so, I would suggest referencing in this section Page 9, Line 42: Why are participants required to be aged between 18-60 years? Justification for this could be provided earlier in the introduction Page 9, Lines 47-49: Provide a reference for these PA guidelines Page 10, Lines 9-11: Will initial screening of cervical cancer patient files at UHL be conducted by the research or treating team? Or are all patients (who may or may not be eligible) from the gynaecological and oncology outpatient departments at UHL being approached by the treating team? I would suggest clarifying this Page 10, Lines 30-42: The reference used is specific to pilot studies, not feasibility (or preliminary) studies. Furthermore, not essential, but would consider using these previous interventional studies to calculate a more accurate sample size. However, it is acknowledged this is not always needed in feasibility studies (and is sometimes even discouraged) Page 10, Lines 49-52: I would suggest capitalising the F in “figure 1” and “figure 2” Page 10, Lines 54-57: This reference is specific to those living beyond ovarian cancer, not gynaecological cancer in general Page 13, Line 9: I am unsure what is meant by “whilst those measures expected to change, based on the literature”. Is this in reference to measures reported in the introduction? In the introduction, you have referred to Quality of Life, so why is this not being assessed at 6 weeks too? It would be good to clarify this Page 14, Line 11: What is the minimum number of days participants must wear the accelerometer (you note it will be a maximum of 8 days)? This will have an impact on your analysis capabilities Page 14, Lines 31-35: Reference/s needed Page 15, Lines 34-41: Who will conduct the semi-structured interview? Please provide details Page 15, Line 41: I would recommend replacing this reference with a peer-reviewed source, rather than a website Page 16, Line 24: “we” should have a capitalised W Page 16, Lines 29-36: Are these values based upon previous work in the oncology field, and/or previous work in your group? Page 17, Lines 8-11: Has the accelerometer analysis protocol been decided yet? If so, I would suggest providing details Discussion: Page 18, Line 8: Delete the word “of” Page 18, Lines 15-22: References needed for this statement Page 18, Line 48: This reference appears to be to a government website. I would amend to the specific government report/guidelines Page 19, Lines 3-6: This reference did not study those living with or beyond cancer, instead it investigated obese women who are otherwise healthy Page 19, Lines 6-10: Did a desire for aerobic training, as opposed to resistance training, arise in the need’s assessment with your cervical cancer population?
--	---

	Figure 1: Page 26: Though basic intervention information is provided in this figure; it could be further improved upon to make it more appealing to readers (e.g., reduce white space) and provide necessary intervention information in an easy-to-read format
--	--

VERSION 1 – AUTHOR RESPONSE

Reviewer comment	Response to reviewer's comment
ABSTRACT	
A definition of physical activity would be desirable	Thank you for this consideration. A definition of Physical activity has been included in the abstract (Page 2, lines 3-4)
More clearly report what is considered to be the primary outcome(s) of interest (feasibility) and what is considered secondary outcome(s) of interest (QoL, anxiety and depression, fatigue etc).	Thank you for suggesting this change in language. We appreciate that this is a common way to report study aims, however, we are not reporting statistical changes in outcomes such as PA and health measures but rather we are reporting the trends in means, and we cannot infer effectiveness from this. Therefore, these will not be referred to as the secondary objectives. We have changed the wording of these objectives in the abstract to enhance clarity of the overall aims (Page 2, Lines 6-8) which is similar to a feasibility trial previously published in BMJ open (Olotu et al., 2019)
The dissemination statement needs to align more clearly with the PPI section, where it is stated that the PPI will involve in 'dissemination of results to the wider community as appropriate'. Please outline any planned outputs resulting from the engagement (not just involvement) with the PPI/E.	Thank you for your feedback. This information in the abstract has been adjusted (Page 2, lines 24-25)
Report how feasibility will be assessed, at what time points and how it will be analysed. Specifically, what are the criteria for concluding feasibility?	Thank you for bringing this to our attention. Specific feasibility criteria are outlined in detail in the statistical analysis section and so it is not deemed necessary to repeat this in the abstract section. The abstract has been altered to clearly state the existence of feasibility criteria and thresholds (Page 2, lines 18-19)
Page 4, Line 22: Spell "UK" in full	Thank you for noting this. It has been amended (Page 2, line 10)

INTRODUCTION	
This section could provide more clarity in relation to the PA guidelines for cancer survivors and the lack of recommendations for cervical cancer survivors, as stated later in the Discussion. It feels that this addition would give a clearer context for this piece of research being timely and much needed.	Thank you for this feedback. Information has been added to support this in the introduction (Page 5, lines 12-14)
Lines 41-48: please review punctuation.	Thank you for bringing this to our attention. Relevant changes have been made (Page 4, line 17)
I would recommend the authors carefully consider sentence length in some of their sections (especially the introduction). This will assist with the flow of their writing and reader comprehension.	Thank you for highlighting this. Sentence length through-out the manuscript has been considered and changed where necessary.
Page 6, Lines 6-9: This statement should be supported by peer-reviewed literature (or specific government reports) relevant to changes in cervical cancer survival rates	Thank you for this reflection. To further support the role played by improved clinical management in increased survival rates, a peer reviewed reference has also been added (Page 4, line 3)
Page 6, Lines 9-13: Reference needed	Thank you for bringing this to our attention. References have been added (Page 4, lines 7-9)
Page 6, Lines 29-36: Reference/s needed to support these side effects	Thank you for noting this. Relevant peer reviewed references have been added (Page 4, lines 11- 13).
Page 6, Lines 54-57: I suggest rewording "reportedly more greatly affected"	Thank you for your suggestion. The wording of this sentence has been changed slightly (Page 4, line 22-24).
Page 7, Line 17: This reference does not refer in any way to "provision of treatment-related morbidity and clinical support services" for cervical cancer. Please replace reference	Thank you for noting this mistake. This reference has been replaced (Page 5, line 7)
Page 7, Lines 36-41: References needed	Thank you for your suggestion. However, main points in this section are referenced adequately and it is not clear as to what information needs further support.
METHODS	
More detail is needed on the 'screening to determine potential participants PA levels will take place via a telephone call with a member of the research team'. What counts as physical activity and over what time period is this being considered?	Thank you for this observation. The Scottish Physical Activity Questionnaire was used to determine participants' PA levels. We were specifically interested to know whether potential participants took part in more than 150 minutes of moderate intensity activity per week ('in the last 7 days') as this is an indicator of eligibility. Additional information has been

	added (Page 9, Lines 18-21)
It would be useful to include 'number of minutes that a participant spends in a moderate intensity activity each day' in the daily intervention diary.	Thank you for this consideration. This would have potentially been a very good addition to the intervention diary. A caveat to including this in the diary is that participants are not provided with an objective measure of moderate intensity physical activity, just steps via the fitbit. During the programme they are educated on what moderate intensity PA would feel like (For e.g. PA which results in increased heart rate and precipitation) along with the general number of steps that one would need to do in 10 minutes. Therefore, self-reported MVPA may be a misleading measure. Perhaps, number of brisk walks per week and the length of these would have been good alternatives.
A number of the weekly diary entries are likely to fluctuate throughout the week/day, what is the rationale for only collecting them weekly?	Thank you for voicing this reflection. The diary does ask participants to write their steps and to rate their mood daily, which will provide participants with the opportunity to self-reflect on their well-being on a daily basis. It is acknowledged that the weekly reflections are wellbeing aspects that may fluctuate through-out the week, however, it was decided that these will be collected weekly in order decrease participant burden and to increase adherence to the diary.
Who will be administering the health coaching sessions? Are these also delivered virtually?	Thank you for your question. The health coaching will be delivered by the study co-ordinator (NM) and these sessions will be delivered online via video call using Microsoft teams or via telephone call where this is most convenient for the participant. This information has been altered (Page 10, Line 20)
Clarity is needed for the following statement: '...whilst those measures expected to change, based on literature, will be taken at week 6'. It is unclear what the author means by this; are all items not expected to change?	Thank you for requesting clarity on this. This sentence has been slightly changed to provide further clarity (Page 12, Lines 3-5). Generally, literature suggests that such psychological constructs take at least 12- weeks to change whereas PA and menopausal symptoms may change within 6-weeks, and therefore, to not over burden participants unnecessarily, we have made the decision to only measure these 2 outcomes at week-6.
It would be beneficial to discuss the outcomes as primary outcomes relating to feasibility and acceptance and secondary outcomes relating to PA and health.	Thank you for your suggestion. As previously mentioned, we cannot refer to PA and health measure outcomes as secondary objectives as we will not be exploring statistically significant changes. Therefore, similar to a previous feasibility trial (Morris et al., 2019), we have decided to use feasibility and acceptability objectives. We have added information to the manuscript to explain what the primary

	outcome measure would be in a pilot trial or definitive main trial (Page 12, line 11-12)
Clarity is needed for the following statement: 'The completion rates of the PA and health measures at each time point will be summarised'. Do the authors mean the number of participants that data is collected for, and the completeness of data or something else?	We would like to thank the reviewer for bringing this confusion to our attention. Your interpretation of this is correct as we mean that the number of participants who complete the PA measures (e.g. wear the monitor) and health measures (complete questionnaires) will be summarised as a reflection of compliance with these measures. This sentence has been altered to reflect this (Page 13, lines 18-19)
'If an eligibility criterion accounts for 40% ineligibility, it will be investigated further' – how? Provide the specific investigation planned.	Thank you for your comment. The relevant changes have been made to add clarity to this point (Page 15, lines 9-10)
What is the rationale for a recruitment rate of 30 participants in the first 3-months as the threshold for feasibility?	Thank you for your inquiry. A recruitment aim of 30 participants in the first 3-months was chosen after consulting with statistics related to the number of cervical cancer diagnoses per year along with the rate of diagnoses at university hospitals of Leicester where recruitment was taking place. Therefore, this rate was chosen as it was a realistic target, and it would also be a rate that would yield an appropriate number of participants for a definitive trial over a longer period of time.
What is meant by '75% of those identified are recruited'? 75% of eligible participants or 75% of potential participants approached?	Thank you for requesting this information. This has been updated to add clarity (Page 15, Line 9)
In regard to feasibility of and compliance with intervention components, what threshold will be used to conclude feasibility?	Thank you for noting this lack of information. Threshold for compliance with intervention components will be 70%, this has been added to the statistical analysis section (Page 15, Lines 14-15)

Collaboration with PPI/E is gold standard in health research; therefore, authors are to be congratulated for the effort. However, the research team does not clearly articulate who was part of the PPI group (patients, carers, members of support groups, members of specific communities, people living beyond cancer etc) Mainly that the PPI members were involved in the assessment of the 'potential burden on participants' and had an active role in informing the intervention content. How were these 'discussions' managed, and in what ways was feedback collected and integrated into the intervention content?	Thank you for your comment. Further information on the PPI group have been included (Page 7, Lines 10). Additional information on the feedback collection methods are detailed in a new table which has been added (Page 8).
Please indicate if a PPI/E protocol exists and where it can be accessed. It would be useful to learn whether a PPI/E plan has been designed to outline the 'involvement' and 'engagement' elements of this collaboration throughout different stages of the project and expected outputs. I would suggest a table describing the different activities that the PPI will contribute to (e.g. how they might get involved in different aspects of this project, their role in commenting on and developing the patient facing documents or other research materials, if there is any involvement in data collection or analysis as user researchers etc).	Thank you for this suggestion. A PPI/E protocol was not established for this research. A table has been inputted which outlines the PPI involvement at different research stages (Page 8).
Authors mention that the PPI/E will be involved in the dissemination of results to the wider community. Authors do state that results will be shared with the University Hospitals of Leicester and Leicester Hospital's charity. Please give examples of planned activities, if any (e.g. science festivals open to the public or debates on research open days where members of the public are invited, awareness of research through media, dissemination to members of the public etc). In addition, has the research team considered involving the PPI/E in the Equality and Health Inequalities Impact Assessment of the intervention (e.g. addressing potential inequalities due to race and ethnicity, learning or physical disabilities, digital exclusion etc)?	Thank you for your insightful and important comments. At the moment, there are plans to share the results of this feasibility trial during a presentation with healthcare providers at the University Hospitals of Leicester and to clinicians at an NCRI workshop. PPI members have suggested that a presentation to all participants in the study would be an effective means to distribute knowledge and also to encourage networks between survivors to be made. Further plans to disseminate research output to the wider community have not yet been finalised

Participants and recruitment: Please provide a justification for the upper age limit.	Thank you for your feedback. The upper age limit of 60 years old has been chosen as cancer statistics suggest that cervical cancer is predominantly a cancer of women between the ages of 30- 34 years. Therefore, to ensure that the programme was tailored as effectively as possible to target the majority of this population, it was deemed necessary to have an upper age limit of 60 years.
The feasibility of inclusion criteria: Please capitalize the first letter of the first word	Thank you for bringing this to our attention. It has been amended (Page 15, line 4)

Page 9, Lines 25-37: Do you have any published work referring to this previous PPI work? If so, I would suggest referencing in this section	Thank you for your suggestion. This PPI work has been previously published since this paper was submitted to BMJ open. It has been referenced in text (Page 7, line 14)
Why are participants required to be aged between 18-60 years? Justification for this could be provided earlier in the introduction	Thank you for your feedback. The upper age limit of 60 years old has been chosen as cancer statistics suggest that cervical cancer is predominantly a cancer of women between the ages of 30- 34 years. Therefore, to ensure that the programme was tailored as effectively as possible to target the majority of this population, it was deemed necessary to have an upper age limit of 60 years. Reference to this in the introduction has been added (Page 5 , lines 24-25; Page 6, lines 1-2)
Page 9, Lines 47-49: Provide a reference for these PA guidelines	Thank you for suggesting this. The relevant reference has been added (Page 9, line 5)
Page 10, Lines 9-11: Will initial screening of cervical cancer patient files at UHL be conducted by the research or treating team? Or are all patients (who may or may not be eligible) from the gynaecological and oncology outpatient departments at UHL being approached by the treating team? I would suggest clarifying this	Thank you for bringing this confusion to our attention. This sentence has been altered to clarify this (Page 9, lines 11-12)
The reference used is specific to pilot studies, not feasibility (or preliminary) studies. Furthermore, not essential, but would consider using these previous interventional studies to calculate a more accurate sample size. However, it is acknowledged this is not always needed in feasibility studies (and is sometimes even discouraged)	Thank you for this remark. Pilot studies and feasibility studies are terms referred to interchangeably within literature, as both refer to preliminary research prior to a definitive trial. The reference used mainly refers to pilot trials but it also refers to feasibility studies. The National Institute for Health Research (NIHR) provide specific guidelines on feasibility trials which we have used to guide many aspects of our trial (for e.g. Justifying-sample-size-for-feasibility-study-updated-22-Feb-2019.pdf). A number of considerations informed the justification of a sample size of 30 participants, including the known incidence of cervical cancer, taking into consideration the inclusion and exclusion criteria; the limited time frame and resources for recruitment; and considering what would be an adequate sample size to collect the information needed (device assessed PA and questionnaire data). The NIHR does not suggest that power calculations are needed for a feasibility trial. This would be an aim of the pilot trial were that to take place

Page 10, Lines 49-52: I would suggest capitalising the F in “figure 1” and “figure 2”	Thank you, this has now been amended. (Page 10, Lines 9-10)
Page 10, Lines 54-57: This reference is specific to those living beyond ovarian cancer, not gynaecological cancer in general	Thank you for this comment. A reference has been added which also supports this statement for gynaecological cancer survivors (Page 10, line 12).
Page 13, Line 9: I am unsure what is meant by “whilst those measures expected to change, based on the literature”. Is this in reference to measures reported in the introduction? In the introduction, you have referred to Quality of Life, so why is this not being assessed at 6 weeks too? It would be good to clarify this	Thank you for requesting clarity on this. This sentence has been slightly changed to provide further clarity (Page 12, Lines 3-5). Generally, literature suggests that such psychological constructs take at least 12- weeks to change whereas PA and menopausal symptoms may change within 6-weeks, and therefore, to not over burden participants unnecessarily, we decided to only measure these 2 outcomes at week-6.
Page 14, Line 11: What is the minimum number of days participants must wear the accelerometer (you note it will be a maximum of 8 days)? This will have an impact on your analysis capabilities	Thank you for this comment. Participants are asked to wear the accelerometer for the full 8 days. Participants who provide data spanning 3 days or More will be included in the analysis. This information has been added (Page 12, line 10-11)
Page 15, Lines 34-41: Who will conduct the semi-structured interview? Please provide details	Thank you for requesting this information. This sentence has been amended. (Page 14, Line 8)
Page 15, Line 41: I would recommend replacing this reference with a peer-reviewed source, rather than a website	Thank you for bringing this to our attention. This reference has been updated (Page 14, line 11)
Page 16, Line 24: “we” should have a capitalised W	Thank you. This has been changed.
DISCUSSION	
Page 16, Lines 29-36: Are these values based upon previous work in the oncology field, and/or previous work in your group?	Thank you for your comment. These values are based on expertise within the research group of conducting feasibility trials.
Page 18, Line 8: Delete the word “of”	Thank you for your comment. The sentence has been rearranged

Page 18, Lines 15-22: References needed for this statement	Thank you for referring to this. A reference has been added (Page 17, line 6)
Page 18, Line 48: This reference appears to be to a government website. I would amend to the specific government report/guidelines	Thank you for this comment, The reference has been updated (Page 17, line 20)
Page 19, Lines 3-6: This reference did not study those living with or beyond cancer, instead it investigated obese women who are otherwise healthy	Thank you for noting this. We have revised all references to ensure that they are correct. This has been replaced with the correct reference (Page 18, line 2)
Page 19, Lines 6-10: Did a desire for aerobic training, as opposed to resistance training, arise in the need's assessment with your cervical cancer population?	Thank you for your consideration. Walking specifically was a preference of participants during the need's assessment study. We have recently published the study outlining the development of this intervention which has been cited in the discussion.
FIGURES	
Page 26: Though basic intervention information is provided in this figure; it could be further improved upon to make it more appealing to readers (e.g., reduce white space) and provide necessary intervention information in an easy-to-read format	Thank you for your insight. The figure has been slightly changed to decrease white space. It is not felt that key intervention information is missing however, as we do not want to repeat information from the intervention table (Page 17, line 24)

VERSION 2 – REVIEW

REVIEWER	Edward Stanhope Staffordshire University
REVIEW RETURNED	24-Sep-2021

GENERAL COMMENTS	This protocol submission titled 'ACCEPTANCE: Protocol for a feasibility study of a multicomponent physical activity intervention following treatment for cervical cancer' has a strong rationale and the methodological features have been well considered. The researchers are commended on their clear and well-structured response to reviewer comments. Abstract: Although Physical Activity has been defined as 'any movement that uses energy', it is unreferenced. I suspect it was paraphrased from Caspersen CJ, Powell KE, Christenson GM: Physical activity, exercise, and physical fitness: definitions and distinctions for health-related research. Public Health Rep. 1985, 100 (2): 126-131. Thank you for addressing all other reviewer comments for this section. Introduction: Authors have indicated that 'exercise recommendations for cervical cancer survivors do not exist, thus supporting the need for tailored PA promotion in this population'. While I accept that specific exercise recommendations for cervical cancer may not exist, there are general PA guidelines for cancer survivors worth acknowledging, including the ACS guidelines (Doyle, C., Kushi, L. H., Byers, T., Courneya, K. S., Demark-Wahnefried, W., Grant, B., ... & Andrews,
---

	K. S. (2006). Nutrition and physical activity during and after cancer treatment: an American Cancer Society guide for informed choices. CA: a cancer journal for clinicians, 56(6), 323-353) (Doyle, C., Kushi, L. H., Byers, T., Courneya, K. S., Demark-Wahnefried, W., Grant, B., ... & Andrews, K. S. (2006). Nutrition and physical activity during and after cancer treatment: an American Cancer Society guide for informed choices. CA: a cancer journal for clinicians, 56(6), 323-353.), the ESSA (Hayes, S. C., Spence, R. R., Galvão, D. A., & Newton, R. U. (2009). Australian Association for Exercise and Sport Science position stand: optimising cancer outcomes through exercise. Journal of science and medicine in sport, 12(4), 428-434.) and ACSM (Schmitz, K. H., Courneya, K. S., Matthews, C., Demark-Wahnefried, W., Galvão, D. A., Pinto, B. M., ... & Schwartz, A. L. (2010). American college of sports medicine roundtable on exercise guidelines for cancer survivors. Medicine & Science in Sports & Exercise, 42(7), 1409-1426.) among others. Thank you for addressing all other reviewer comments for this section. Methods: Thank you for clarifying the screening process and that the Scottish Physical Activity Questionnaire will be used to determine PA levels over the last 7 days. Thank you for considering the addition of 'number of minutes that a participant spends in a moderate intensity activity each day' in the daily intervention diary. The researchers raise a valid reason why this is not possible. The researchers may find the following paper useful when considering stride count thresholds for classifying MVPA using Fitbit technology; Silva, G. S., Yang, H., Collins, J. E., & Losina, E. (2019). Validating Fitbit for Evaluation of Physical Activity in Patients with Knee Osteoarthritis: Do Thresholds Matter?. ACR open rheumatology, 1(9), 585-592. Thank you for clarifying that participants will write their steps and rate their mood daily and the reason for weekly reflections is due to patient burden concerns. Thank you for confirming that the health coaching sessions will be delivered online. I think it is worth specifically noting in text that these are conducted by the study co-ordinator. Thank you for confirming that 'A recruitment aim of 30 participants in the first 3-months was chosen after consulting with statistics related to the number of cervical cancer diagnoses per year along with the rate of diagnoses at university hospitals of Leicester where recruitment was taking place. Therefore, this rate was chosen as it was a realistic target, and it would also be a rate that would yield an appropriate number of participants for a definitive trial over a longer period of time', I think this justification is worth including in your protocol for complete transparency. Whilst the researchers have stipulated that 'the upper age limit of 60 years old has been chosen as cancer statistics suggest that cervical cancer is predominantly a cancer of women between the ages of 30-34 years. Therefore, to ensure that the programme was tailored as effectively as possible to target the majority of this population, it was deemed necessary to have an upper age limit of 60 years' the question remains as to why 55 years old or 50 years old etc. was not suitable. The decision and justification to limit to <60 years could be more clearly reported. Thank you for addressing all other reviewer comments for this section. Discussion: Thank you for addressing all reviewer comments for this section.
--	---

Reviewer comment	Response to reviewer's comment *All page and line numbers correspond to the manuscript with marked tracked changes
--

VERSION 2 – AUTHOR RESPONSE

Reviewer comment	Response to reviewer's comment *All page and line numbers correspond to the manuscript with marked tracked changes
ABSTRACT	
Although Physical Activity has been defined as 'any movement that uses energy', it is unreferenced. I suspect it was paraphrased from Caspersen CJ, Powell KE, Christenson GM: Physical activity, exercise, and physical fitness: definitions and distinctions for health-related research. Public Health Rep. 1985, 100 (2): 126-131	Thank you for your feedback. As it is not commonplace to reference in the abstract, the definition of physical activity has been moved to the introduction and includes the full definition as cited in Caspersen, Powell & Christenson (1985; page 5, lines 7-8). Physical activity is explained as 'bodily movement' in the abstract to maintain the succinct flow of language in this section. (Page 2, Line 3)
INTRODUCTION	
Authors have indicated that 'exercise recommendations for cervical cancer survivors do not exist, thus supporting the need for tailored PA promotion in this population'. While I accept that specific exercise recommendations for cervical cancer may not exist, there are general PA guidelines for cancer survivors worth acknowledging, including the ACS guidelines (Doyle, C., Kushi, L. H., Byers, T., Courneya, K. S., Demark-Wahnefried, W., Grant, B., ... & Andrews, K. S. (2006). Nutrition and physical activity during and after cancer treatment: an American Cancer Society guide for informed choices. CA: a cancer journal for clinicians, 56(6), 323-353) (Doyle, C., Kushi, L. H., Byers, T., Courneya, K. S., Demark-Wahnefried, W., Grant, B., ... & Andrews, K. S. (2006). Nutrition and physical activity during and after cancer treatment: an American Cancer Society guide for informed choices. CA: a cancer journal for clinicians, 56(6), 323-353.), the ESSA (Hayes, S. C., Spence, R. R., Galvão, D. A., & Newton, R. U. (2009). Australian Association for Exercise and Sport Science position stand: optimising cancer outcomes through exercise. Journal of science and medicine in sport, 12(4), 428-434.) and ACSM (Schmitz, K. H., Courneya, K. S., Matthews, C., Demark-Wahnefried, W., Galvão, D. A., Pinto, B. M., ... & Schwartz, A. L. (2010). American college of sports medicine roundtable on exercise guidelines for cancer survivors. Medicine & Science in Sports & Exercise, 42(7), 1409-1426.) among others.	Thank you for bringing this point to our attention. We agree that the exercise guidelines for cancer survivors more generally should be referred to. The ACSM guidelines have been cited in the introduction (Page 5, Lines 14-16)
METHODS	
Thank you for confirming that the health coaching sessions will be delivered online. I think it is worth specifically noting in text that these are conducted by the study co-ordinator.	Thank you for your contribution. This has been altered in the methods section (Page 10, line 22)

Thank you for confirming that 'A recruitment aim of 30 participants in the first 3-months was chosen after consulting with statistics related to the number of cervical cancer diagnoses per year along with the rate of diagnoses at university hospitals of Leicester where recruitment was taking place. Therefore, this rate was chosen as it was a realistic target, and it would also be a rate that would yield an appropriate number of participants for a definitive trial over a longer period of time', I think this justification is worth including in your protocol for complete transparency.	Thank you for suggesting that this important information is included in the protocol. The methods section has been altered to reflect this (Page 10, lines 2-4).
Whilst the researchers have stipulated that 'the upper age limit of 60 years old has been chosen as cancer statistics suggest that cervical cancer is predominantly a cancer of women between the ages of 30- 34 years. Therefore, to ensure that the programme was tailored as effectively as possible to target the majority of this population, it was deemed necessary to have an upper age limit of 60 years' the question remains as to why 55 years old or 50 years old etc. was not suitable. The decision and justification to limit to <60 years could be more clearly reported.	Thank you for requesting further clarity on this. In addition to our previous comment, there are a number of reasons why the age limit of 18-60 years is in place for recruitment of study participants. Firstly, reaching the national physical activity guidelines of 150-300 minutes of physical activity is a key aim of this programme. These guidelines are recommended for adults ages between 18-65 years. Secondly, the protocol details a largely technology-based programme. Research suggests that whilst fitbit use is accepted amongst over 65s (Rossi et al., 2018), engagement with the internet and technologies is substantially lower amongst older adults (Smith, 2014). Additionally, the majority of research supporting the adoption of technologies in older adult cancer survivors include populations of 60 years and below (e.g. Pope et al., 2019). Additionally, older adults may be more susceptible to comorbidities unrelated and additional to their cancer diagnoses which may influence their ability to engage with and adhere to the physical activity programme. As this is a feasibility trial with limited resources for recruitment and funding for materials, it was deemed important to have this age limit in place to ensure that those recruited would be in a position to adequately test the feasibility and acceptability of the programme. We acknowledge that employing this age range may influence the acceptability of this programme and so we agree that transparency regarding this is important. This has been included in text (Page 9, lines 6-8)